# The Role of Artificial Intelligence in Colorectal Cancer Screening: Lesion Detection and Lesion Characterization

**DOI:** 10.3390/cancers15215126

**Published:** 2023-10-24

**Authors:** Edward Young, Louisa Edwards, Rajvinder Singh

**Affiliations:** 1Faculty of Health and Medical Sciences, University of Adelaide, Lyell McEwin Hospital, Haydown Rd, Elizabeth Vale, SA 5112, Australia; 2Faculty of Health and Medical Sciences, University of Adelaide, Queen Elizabeth Hospital, Port Rd, Woodville South, SA 5011, Australia

**Keywords:** colonoscopy, artificial intelligence, polyp, adenoma, colorectal cancer

## Abstract

**Simple Summary:**

There has been an exponential rise in the availability of artificial intelligence systems in endoscopy in recent years. As a result, maintaining an informed understanding of the utility and efficacy of existing systems has become increasingly complex. This review aims to summarise the expanse of research in this area to guide proceduralists in making informed decisions regarding the use of artificial intelligence in colonoscopy. It focuses primarily on the application of artificial intelligence for the detection and characterisation of colorectal polyps in order to improve the efficacy of colorectal cancer screening and prevention.

**Abstract:**

Colorectal cancer remains a leading cause of cancer-related morbidity and mortality worldwide, despite the widespread uptake of population surveillance strategies. This is in part due to the persistent development of ‘interval colorectal cancers’, where patients develop colorectal cancer despite appropriate surveillance intervals, implying pre-malignant polyps were not resected at a prior colonoscopy. Multiple techniques have been developed to improve the sensitivity and accuracy of lesion detection and characterisation in an effort to improve the efficacy of colorectal cancer screening, thereby reducing the incidence of interval colorectal cancers. This article presents a comprehensive review of the transformative role of artificial intelligence (AI), which has recently emerged as one such solution for improving the quality of screening and surveillance colonoscopy. Firstly, AI-driven algorithms demonstrate remarkable potential in addressing the challenge of overlooked polyps, particularly polyp subtypes infamous for escaping human detection because of their inconspicuous appearance. Secondly, AI empowers gastroenterologists without exhaustive training in advanced mucosal imaging to characterise polyps with accuracy similar to that of expert interventionalists, reducing the dependence on pathologic evaluation and guiding appropriate resection techniques or referrals for more complex resections. AI in colonoscopy holds the potential to advance the detection and characterisation of polyps, addressing current limitations and improving patient outcomes. The integration of AI technologies into routine colonoscopy represents a promising step towards more effective colorectal cancer screening and prevention.

## 1. Introduction

In the context of modern healthcare, the integration of artificial intelligence (AI) has emerged as a transformative force, revolutionising various aspects of medical practice [1]. One promising application lies in the domain of colorectal cancer (CRC), where AI holds the potential to enhance the accuracy and efficiency of polyp detection and characterisation during colonoscopy—a pivotal procedure for early diagnosis and prevention. This review article delves into the dynamic intersection of AI and CRC management, with a specific focus on its application for polyp detection and characterisation during colonoscopy.

CRC accounts for more than 10% of cancer diagnoses and more than 9% of cancer-related mortality worldwide, necessitating effective screening and diagnostic strategies to curb its impact [2]. There is now compelling evidence that the implementation of population CRC screening in developed countries has led to a considerable reduction in its incidence and mortality [3,4]. Colonoscopy serves as the gold standard for both the detection and prevention of CRC, yet its efficacy is contingent on the skill and vigilance of the endoscopist [5]. Despite advances in endoscopic technology and improvement in adenoma detection, adenoma miss rates still remain as high as 26% in tandem colonoscopy studies [6]. Miss rates are particularly high for sessile serrated lesions (SSLs) (27%), proximal advanced adenomas (14%), and flat adenomas (34%) [6]. The integration of AI into colonoscopy holds the promise of augmenting human expertise, potentially reducing the miss rates of these inconspicuous polyps and thereby improving patient outcomes.

Drawing upon a plethora of studies, this review aims to dissect the methodologies and technological advancements that underpin AI-driven polyp detection and characterisation systems, with a particular focus on more recent real-world experiences with AI. By exploring the evolution, challenges, and outcomes associated with these technologies, we strive to provide insights into their potential to reshape CRC management paradigms. While this is not a formal systematic review, it has been based largely on a structured examination of published literature from Pubmed and Embase, with abstracts screened for relevance and reference lists reviewed for additional relevant studies.

## 2. Artificial Intelligence in Colonoscopy

Research in the field of AI-assisted colonoscopy has expanded exponentially in the last 5 years, with a wide range of AI systems now commercially available (Table 1). As a result, understanding the efficacy and accuracy of these individual systems has become increasingly complex while there are limited data available for direct comparison and no form of standardisation exists. Nevertheless, proponents of AI argue that the sophistication of deep learning and the vast datasets on which these systems are trained result in consistent accuracy at a high level. In the absence of standardisation, this review seeks to analyse the efficacy of the commercially available systems and the accuracy of this assertion.

Machine learning involves the development of an algorithm based on a training dataset in order to predict the same pattern in unseen data. Initially, AI systems in endoscopy involved the manual introduction of polyp features to the machine learning algorithm for the program to recognise polyps; however, the accuracy of AI systems has catapulted with the introduction of deep learning. Deep learning is a type of machine learning characterised by self-learning, in that the program extracts data and recognises key features across multiple layers without any requirement for human input. It involves neural networks, imitating the complex interconnected networks of the human brain in order to analyse multiple increasingly complex layers of images. Convolutional neural networks (CNNs) are based on the principle of the visual cortex of the human brain for image processing. Using multiple filters, the CNN extracts key features from multiple versions of the same image before pooling layers to provide a final classification as the output based on learned polyp features. The key advantage of these systems is that the more data that is fed into the system, the more sophisticated the algorithm becomes, as the system is capable of continued independent learning. CNNs are a popular method for image recognition as they offer efficient performance, allowing for use in real-time video applications [7,8].

The number of AI systems developed or in development for upper and lower gastrointestinal endoscopy has expanded exponentially in recent years. Computer-aided detection (CADe) systems recognise characteristic features in order to discern the presence of a polyp within a still image or video. More recently, these systems have been integrated into real-time colonoscopy, alerting proceduralists to the presence of a polyp either with a coloured box around the entire display or a box around the polyp itself. Computer-aided diagnosis (CADx) systems are able to distinguish between polyp types and degrees of dysplasia, from benign hyperplastic polyps to advanced cancers, providing a real-time diagnosis to the proceduralist.

## 3. Polyp Detection

Since 2016, researchers have published deep learning algorithms for polyp detection (CADe) that have been tested in pre-clinical applications, such as polyp detection in still images or videos [9]. Only 3 years later, the first randomised controlled trials (RCTs) comparing CADe with existing standards were published [10]. Since then, there has been a vast amount of research published on real-time CADe systems, with strong support for their efficacy in polyp detection. Of the 15 RCTs reviewed here, 10 demonstrated a statistically significant increase in adenoma detection, although baseline and CADe adenoma detection rates (ADRs) are highly varied because of differing populations and study designs (Table 2) [10,11,12,13,14,15,16,17,18,19,20,21,22,23,24]. Although overall lesion detection is generally improved, many of these systems have been criticised for a lack of impact on the detection of advanced adenomas of heightened clinical significance. Many argue that these larger polyps are less likely to be missed by endoscopists, making the implementation of CADe systems less pivotal. While it may be true that larger polyps are less likely to be missed by endoscopists, the lack of demonstrable impact of CADe systems for advanced adenomas may simply reflect their reduced prevalence and, hence, the larger numbers required to adequately power these studies. For example, in the largest RCT by Xu et al., including 3059 patients, there was a statically significant increase in advanced adenoma (>10 mm, villous component or high-grade dysplasia) detection in the CADe group versus the control group (6.6% vs. 4.9%, *p* = 0.041) [12].

In an effort to synthesise the expanse of research in this area, multiple meta-analyses have been published comparing CADe with high-definition white light imaging (HD-WLI) control groups (Table 3). These studies have universally found an increase in ADR with CADe, with a 1.43–1.78 times increase in ADR versus HD-WLI [25,26,27,28,29,30,31,32,33,34,35]. The most significant difference has been in the detection of diminutive (<5 mm) adenomas. For larger polyps, the results have been varied, with four of the seven meta-analyses specifically analysing >10 mm adenomas finding a statistically significant improvement in detection. Interestingly, in their 2021 meta-analysis, Zhang et al. actually reported a reduction in the detection of advanced adenomas with CADe [34]. While this raises the possibility that the time and concentration consumed by higher diminutive polyp detection with CADe may detract from the detection of advanced lesions, this has not been borne out in other meta-analyses and was not the case in the largest RCT to date [12]. Sessile serrated lesions (SSLs) are a polyp subtype prone to being missed during colonoscopy because of their inconspicuous nature, as they are generally flat and difficult to differentiate from surrounding normal mucosa. For SSLs, RCTs have not been powered to demonstrate an effect as their incidence is considerably lower compared with adenomas. However, three meta-analyses assessed SSLs specifically, demonstrating a between 1.37- and 1.52-times increase in SSL detection with CADe, though one of these did not reach statistical significance [25,30,33].

Overall, prospective studies into CADe for adenoma detection have been optimistic. Although many studies have not shown improved advanced adenoma detection, multiple meta-analyses and the largest RCT to date suggest that this is likely the case, and it has been conclusively demonstrated to improve the detection of diminutive adenomas. However, with the advent of commercially available CADe systems, data are now available in a real-world context, which may have greater generalisability than those conducted in a clinical trial setting. The largest of these, published by Ladabaum et al. in 2023, was a pragmatic real-world retrospective study whereby data were collected following the implementation of CADe in a single centre, compared with concurrent and historical controls [36]. In this study, the introduction of CADe resulted in no statistically significant difference in any detection metric, including ADR, adenomas per colonoscopy, or advanced adenoma detection. This was further supported by Levy et al., who demonstrated a reduction in ADR from 35.2% to 30.3% (*p* < 0.001) in their single-centre cohort study [37]. These studies highlighted the potential pitfalls of the use of CADe, including less thorough mucosal exposure due to a ‘false sense of security’ from the AI assistance; proceduralists dismissing lesions not highlighted by AI; and the cumulative effect of false positive detection and the resulting increase in withdrawal time. However, in two other large real-world propensity score-matched studies including a cumulative 2262 patients following the implementation of CADe, its introduction resulted in a 1.32–1.59-times higher ADR when compared with HD-WLI [38,39].

The differing results in these real-world implementation studies may relate in part to differences in the impact of AI on expert referral centres with already high ADR versus lower ADR proceduralists. Given the nature of the limited availability of CADe systems thus far, few studies have examined their impact on low-ADR endoscopists. As can be seen in Table 2, of the five studies not demonstrating a difference in ADR with CADe, only one study had a baseline ADR of less than 36% [38,39]. In this study by Wang et al., the control group included a second observer and was, therefore, not strictly a ‘standard of care’ control [13]. In one such study with a low baseline ADR, adenoma detection improved from 19.9% to 26.4% with the introduction of CADe [38]. Interestingly, in this study, proceduralists were stratified by experience, with experts defined as having performed more than 1000 colonoscopies, rather than by ADR. In doing so, they found no improvement in ADR in the ‘non-expert’ group. This raises the possibility that baseline ADR is of greater significance than procedural experience when determining the impact of CADe. This was also supported by Repici et al., who compared ADR with and without CADe across 660 colonoscopies performed by non-experts (<2000 colonoscopies) and found no correlation between examiner experience and the impact of AI on ADR [17]. In contrast, although not a controlled comparative study, Biscaglia et al. showed that with the assistance of CADe, trainee endoscopists (200–400 previous colonoscopies) could achieve the same ADR on tandem colonoscopy with expert, high-ADR endoscopists without AI assistance [40]. To the best of our knowledge, no studies have been published to date with stratification between endoscopists on baseline ADR in order to investigate this further.

While ADR is often used as a surrogate marker, the adenoma miss rate (AMR) is the most direct correlate with the potential for bowel cancer development despite surveillance colonoscopy. Few studies have directly examined the impact of CADe in this context. AMR refers to the number of adenomas ‘missed’ during a colonoscopy, generally based on tandem colonoscopy studies where an immediate repeat procedure detects additional adenomas. Three tandem colonoscopy studies (Table 4) have compared AMR for CADe versus HD-WLI, with a significant reduction when using CADe [41,42,43]. The SSL miss rate was higher in all three studies with HD-WLI, with two reaching statistical significance. In addition, non-polypoid and right-sided adenomas, both of which are frequently missed at colonoscopy, were less likely to be missed with the use of CADe. These are promising data for the potential of CADe to standardise the quality of colonoscopy by reducing miss rates for these more inconspicuous polyp subtypes.

Multiple previous studies have demonstrated the impact of fatigue on ADR, presumably because of a higher likelihood of human error. A 2009 retrospective study of 3619 colonoscopies found an ADR of 29.3% in the morning versus 25.3% in the afternoon (*p* = 0.008) [44]. This was reinforced by a prospective study that found that 27% more polyps were detected per patient during early morning cases, with an hour-by-hour decrease in adenoma detection as the day progressed [45]. Given CADe aims to reduce the likelihood of human error, two studies have assessed its role in preventing deterioration in ADR from physician fatigue. Lu et al. undertook a post hoc analysis of two prospective RCTs comparing CADe with HD-WLI, finding that while the ADR in morning sessions was higher in the control group, there was no longer any statistically significant difference in the CADe group [46]. In this cohort, the OR for adenoma detection during afternoon colonoscopy with CADe assistance versus without was 3.81 (95% CI 2.1–6.91) [46]. Similarly, Ritcher et al. performed a retrospective database analysis comparing ADR with CADe versus HD-WLI over the course of a day, demonstrating that while there was a statistically significant trend towards reduction in ADR throughout the day with HD-WLI (*p* = 0.015), this trend was no longer present in the CADe-assisted group (*p* = 0.65) [47].

### 3.1. Criticisms of CADe

The two main criticisms of CADe are the impact on procedure time and the high rates of distracting false positive polyp identifications. In a 2022 ESGE position statement, the overwhelming consensus was that, for the use of CADe to become widespread, it would need to have an acceptable false-positive rate such that it does not significantly prolong procedure times [48].

Despite initial concerns from image- and video-based studies, the actual rates of false positives that have a meaningful impact on withdrawal time appear to be low, with 91% of false positives lasting less than half a second [49]. In their post hoc analysis of an RCT, Hassan et al. found that while overall false positive rates are high (27.3 per colonoscopy), only 5.7% of false positives required an additional exploration time of 4.8 s per false positive, adding a negligible 1% increase in total withdrawal time [50]. Nevertheless, although the majority of false positives are short-lived, they still have a considerable impact on proceduralist fatigue, with more than 80% of gastroenterologists reporting concerns regarding excessive false positive alerts in a 2023 survey assessing one commercially available CADe system [51]. These false positive alerts from CADe are most often related to bubbles or faeces falsely identified as polyps. As a result, Tang et al. examined whether this could be minimised using water exchange colonoscopy (where water is used rather than CO_2_ insufflation during colonoscope insertion while, at the same time, fluid is suctioned to clear the lumen) in order to clear the field of view of the mucosa. In their 2022 study, they demonstrated a significant increase in the additional polyp detection rate with CADe versus HD-WLI after water exchange colonoscopy (30.1% vs. 12.3%, *p* = 0.001), with a lower rate of false positives related to faeces (*p* = 0.007) and bubbles (*p* = 0.001) due to the clearer field upon colonoscope withdrawal [52]. Techniques such as water exchange colonoscopy, therefore, stand to enhance the performance of CADe not only by improving mucosal visualisation but also by reducing rates of distracting false positives.

Regarding withdrawal times, it remains difficult to assess the true mucosal inspection time without this being impacted by the additional time spent on polyp assessment and resection. Though studies generally pause a stopwatch at the time of polypectomy, there are still delays when a polyp is found, for example, while the stopwatch is paused and restarted on each occasion. The most accurate assessment is, therefore, in the withdrawal time in patients where no polyps are found. Of the four meta-analyses from Table 3 directly examining withdrawal time, no study found any significant difference in withdrawal time in patients with no polyps, while three out of four found a slightly longer withdrawal time (up to a mean of 0.46 min) overall with CADe [25,27,29,33]. In all likelihood, despite false positives from CADe, the only meaningful difference in withdrawal times is in the impact on polyp detection.

### 3.2. Cost Effectiveness

There are controversies surrounding the cost-efficacy of implementing CADe-assisted colonoscopy in screening programs. Initially, the increase in adenoma detection will result in an increased healthcare burden because of requirements for pathological evaluation and a shortening of surveillance intervals. However, eventually, the reduction in adenoma miss rates may mean that surveillance guidelines are able to be adjusted, and there are significant cost savings if advanced colorectal cancers are able to be prevented. In 2022, Mori et al. investigated this further by performing a pooled analysis of RCTs, demonstrating that the proportion of patients who were recommended more intensive surveillance according to US guidelines increased from 8.4% in the control group to 11.3% in the CADe group (RR 1.35, 95% CI 1.16–1.57), which would place a significant burden on a strained healthcare system [53]. However, Areia et al. developed a microsimulation model in a hypothetic cohort to show that the implementation of CADe detection in a US population resulted in a yearly additional prevention of 7194 colorectal cancer cases and 2089 related deaths, with cost savings of USD 290 million [54]. This is aptly described in the World Endoscopy Organisation position statement on AI in colonoscopy in 2023, which states the following: ‘In the short term, use of CADe is likely to increase health-care costs by detecting more adenomas’, but ‘the increased cost by CADe could be balanced by savings in costs related to cancer treatment due to CADe-related cancer prevention‘ [55].

### 3.3. Summary

CADe systems lead to improved adenoma detection, particularly for diminutive adenomas and polyp subgroups more likely to be missed because of human error, including non-polypoid adenomas, right-sided adenomas, and SSLs. While this has not yet been consistently supported by ‘real-world’ studies, the existing retrospective studies introduce forms of bias that may influence results. What has been demonstrated, however, is that, with the support of CADe, regular endoscopists can achieve equivalent performance in adenoma detection to expert high-ADR endoscopists in referral centres, standardising the quality of service provision. Given the dramatic increase in demand for colonoscopy with the implementation of population screening programs, not all patients will have access to expert referral centres for colonoscopy. CADe systems, therefore, have the capacity to make equality of healthcare provision a reality despite inevitable resource limitations. This sentiment is echoed by the European Society of Gastrointestinal Endoscopy (ESGE) 2022 position paper on AI in gastrointestinal endoscopy, stating that ‘the task of AI is to lift the less experienced to the level of experienced endoscopists rather than to further increase the high ADR values of the high-detector experts’ [48]. In this way, CADe is clearly meeting its objective.

## 4. Polyp Characterisation

In addition to lesion detection, the other primary focus of AI systems in colonoscopy has been on the characterisation of polyps (computer-aided diagnosis—CADx). Although expert interventional endoscopists with advanced mucosal imaging are able to achieve a high degree of accuracy in histology prediction, this requires specialised training, experience, and time that may not be available in the general endoscopy setting [56]. Accurate histology prediction is of particular importance in two commonly encountered settings in colonoscopy. For diminutive (<5 mm) polyps, accurate prediction facilitates the safe use of the ‘resect and discard’ and ‘do not resect’ strategies, as discussed below [57]. For larger polyps, the prediction of histology guides appropriate referral pathways for non-interventional endoscopists, either for endoscopic or surgical resection.

For overall histology prediction, multiple image-based studies and three meta-analyses have demonstrated the superiority of CADx compared with non-expert endoscopists [58,59,60,61,62,63,64,65,66,67,68]. However, in each of these meta-analyses, CADx has been unable to outperform expert endoscopists [58,59,60]. In addition, in existing real-time colonoscopy studies, CADx has not been shown to significantly improve the sensitivity or specificity of overall histology prediction. Barua et al. compared CADx with non-expert endoscopists (1–5 years of colonoscopy experience) across 518 patients with 892 polyps and demonstrated no significant difference in sensitivity (90.4% vs. 88.4%) or specificity (85.9% vs. 83.1%) [69]. When compared with expert endoscopists, Li et al. found CADx to be inferior in terms of both sensitivity (61.8% vs. 70.3%, *p* < 0.001) and overall accuracy (71.6% vs. 75.2%, *p* = 0.023) [70].

### 4.1. Diminutive Polyps

Despite a degree of variability in the evidence described above, there are certain circumstances where the accuracy of CADx has been more clearly established, including for the diagnosis of diminutive polyps. In this context, accurate histology prediction serves to avoid unnecessary and expensive pathologic evaluations. The Preservation and Incorporation of Valuable Endoscopic Innovations (PIVI) initiative is a program from the American Society for Gastrointestinal Endoscopy (ASGE) aiming to establish thresholds for endoscopic technologies aimed at addressing important clinical questions and needs in endoscopic diagnosis and intervention [57]. A key focus has been on two strategies to reduce the burden of the histopathological analysis of diminutive colorectal polyps. According to PIVI, diminutive polyps outside of the rectosigmoid colon should be resected but do not require pathological analysis provided endoscopic imaging-based histology prediction results in more than 90% agreement with pathology for surveillance intervals (the ‘resect and discard’ strategy). In addition, diminutive rectosigmoid polyps do not require resection if the endoscopic appearance is of a hyperplastic polyp, provided endoscopic imaging achieves a negative predictive value of more than 90% for adenomatous histology (the ‘do not resect’ strategy). In this context, CADx has been able to comprehensively surpass expectations.

Multiple image-based studies have shown CADx to be superior to non-expert endoscopists for diminutive polyps, with a 96–97% NPV and a sensitivity of 92.3–98.1% [71,72,73,74,75]. Once again, the accuracy of CADx has not outperformed expert endoscopists; however, the widespread adoption of CADx would allow endoscopists of all levels of expertise to employ the ‘do not resect’ or ‘resect and discard’ strategies, thereby improving the cost-effectiveness of colonoscopic screening programs. This was assessed in real-time colonoscopy by Rondonotti et al., including all patients with at least one diminutive rectosigmoid polyp assessed by an endoscopist with CADx assistance [76]. An AI-assisted high-confidence prediction was made in 92.3% of polyps, with NPVs of 91% and 97.4% agreement with ESGE surveillance intervals. Although the initial AI-assisted accuracy was significantly higher in expert (91.9%) versus non-expert (82.3%) endoscopists, there was a significant trend over time in non-experts, such that, for the final 50 polyps, there was no difference in NPV for non-experts (95.2%) versus experts (93.9%).

In fact, certain studies have argued that, for diminutive polyps, pathologic analysis can be misleading, and CADx systems may even outperform the gold standard. In 2019, Ponugoti et al. highlighted the significant discordance that exists between high-confidence expert endoscopist histology prediction and pathologic evaluation for ≤3 mm polyps, postulating that, for polyps of this size, there are frequently issues with processing and retrieval [77]. Subsequently, Shahidi et al. examined the accuracy of CADx diagnoses of 644 ≤3 mm polyps, with a discrepancy between endoscopic and pathological diagnoses in 28.9% of lesions [78]. CADx agreed with expert endoscopists in 90.3% of discordant cases, again highlighting the potential inaccuracy of pathology as the accepted gold standard for polyps of this size.

Critics of CADx argue that the histological predictions of these systems are significantly influenced by the dataset on which they are trained. For example, in datasets with an under-representation of SSLs, the CADx system may be less likely to report a lesion as such. To assess the consistency of these systems, Hassan et al. compared the histology predictions of two CADx systems trained on differing datasets: CAD-EYE and GI-Genius [79]. They found no difference in sensitivity or specificity for the two systems. For ≤5 mm rectosigmoid polyps, the negative predictive value well surpassed the PIVI threshold for both the CAD-EYE (97%) and GI-Genius (97.7%) systems. Based on the ESGE surveillance guidelines, there was 98.3% agreement with guideline-recommended surveillance intervals with both systems. While datasets may impact the outputs of these systems, it is likely that the high volume of polyp images in the training sets is such that the accuracy is more than adequate to facilitate widespread use of the ‘resect and discard’ and ‘do not resect’ strategies.

### 4.2. Larger Polyps

For larger polyps, the potential benefit of CADx is in the identification of appropriate resection strategies or appropriate referral in the case of non-interventional endoscopists. Three studies have examined CADx specifically in larger polyps in comparison with endoscopists. Luo et al. trained a CADx system and tested this on a 1634-image dataset from 156 lesions with high-grade dysplasia or adenocarcinoma [80]. The polyps were stratified by the CADx system into ‘P0’ with a submucosal invasion depth of less than 1000 µm and, therefore, endoscopically resectable or ‘P1’ where there was at least deep submucosal invasion or more advanced cancer. In the testing set, the model had an overall accuracy of 91.1%, a sensitivity of 91.2%, and a specificity of 91.0%, with no significant difference in accuracy compared with experienced interventional endoscopists. When only early adenocarcinomas were included in the analysis, the CADx model was superior to experienced endoscopists (sensitivity 65.3% vs. 40.0%) for differentiating endoscopically resectable lesions, suggesting there may be surface signatures on polyps even with deep submucosal invasion that have not yet been identified by experts in advanced mucosal imaging. Nemoto et al. analysed 1513 early adenocarcinomas, from intramucosal to deep submucosal invasive cancer, comparing their CADx system with trainee and expert endoscopists [81]. CADx showed high specificity at 94.4% for deep submucosal invasion, although sensitivity was low at 59.8%. The AUROC was 85.1% and was equivalent to the two experts (88.2% and 85.9%) and superior to the trainees (77%, *p* = 0.0076 and 66.2%, *p* < 0.001). Yao et al. developed a CADx system trained on 339 large sessile polyps, differentiating malignant from non-malignant polyps [82]. The overall accuracy was 90.4%, which was comparable to expert endoscopists and superior to both senior and junior endoscopists [82]. In this study, with the assistance of CADx, the accuracy of junior endoscopists improved from 75.4% to 85.3% (*p* = 0.002).

While CADx systems are yet to convincingly outperform expert endoscopists in guiding resection strategies, the future of these systems may be in optimising appropriate referrals to experts in endoscopic resection. Additionally, they may obviate the need for a biopsy prior to referral. This is of particular importance as biopsies have been well established as a strong predictor of failed en bloc endoscopic submucosal dissection for colorectal polyps, increasing the odds of severe fibrosis by more than eight times [83].

For expert interventionalists, one role of CADx may be in combination with endocytoscopy systems. Endocytoscopy involves a device that can be either incorporated into the endoscope or as a separate probe-based system, utilising a high-power fixed-focus lens to achieve ultra-high magnification in excess of 450× [84]. This novel technology allows for in vivo visualisations of tissue at the cellular level in real time, with accuracy as high as 85.8–97% for detecting the depth of submucosal invasion [85,86,87,88]. However, these systems require significant training and experience to interpret images. This technology may become more accessible with the advent of AI systems, with EndoBRAIN and EndoBRAIN-Plus now commercially available for the interpretation of endocytoscopic images. Studies thus far have demonstrated a high degree of accuracy for endocytoscopy-based CADx systems, with specificity of up to 97.3–98.9% for differentiating invasive cancer from non-malignant adenoma [89,90]. Kudo et al. compared AI with both trainee and expert endoscopists for endocytoscopic interpretation, with superior accuracy (98% vs. 69% and 93.3%, *p* < 0.001), sensitivity (96.9% vs. 70.8% and 92.8%, *p* < 0.001), and specificity (100% vs. 65.7% and 94.3%, *p* < 0.001) [91]. While these studies demonstrate some benefit for even expert endoscopists in differentiating invasive cancers from non-malignant adenomas, the eventual goal of CADx with endocytoscopy would be to differentiate between depths of submucosal invasion in order to assess suitability for endoscopic resection techniques, a feat not able to be consistently achieved by even the most experienced interventionalists.

In addition, another area for further study that may impact expert endoscopists would be in the assessment of resection margins. To date, no endoscopic systems have been developed for this purpose; however, a recent study performed using hyperspectral imaging on surgical specimens showed high accuracy (AUC 97%) for classifying the components of resected tissue into cancer, adenomatous margins, and healthy mucosa [92]. While this is essentially a proof-of-concept study only, it has highlighted the potential for AI to analyse the completeness of large resections and, therefore, theoretically reduce adenoma recurrence rates.

### 4.3. Summary

CADx systems have been proven to be highly accurate in differentiating neoplastic from non-neoplastic polyps, as well as in recognising invasive cancers. Similar to CADe, these systems are yet to consistently outperform expert endoscopists. Nevertheless, their future may be in the elevation of the accuracy of regular endoscopists to nearing that of highly trained interventionalists in order to guide conservative strategies for diminutive polyps and appropriate referral strategies for larger polyps requiring advanced resection techniques.

## 5. Conclusions

This review provides compelling evidence of the transformative potential of artificial intelligence in the realm of polyp detection and characterisation during colonoscopy. The key findings underscore two crucial aspects that significantly impact healthcare provision, particularly in resource-constrained settings.

First and foremost, the evidence reviewed demonstrates that CADe enhances adenoma detection in studies with low baseline ADR and increases the detection of inconspicuous polyps more frequently missed by endoscopists. This outcome carries substantial implications for public health, as it promises to bolster the consistency of healthcare delivery. In regions or communities where access to highly trained interventionalists may be limited, AI can serve as a reliable and consistent ally in early polyp detection, potentially preventing the progression of colorectal cancer and improving patient outcomes. This democratisation of expertise through AI could bridge the gap in healthcare equality, ensuring that more individuals receive accurate and timely diagnoses, ultimately reducing the burden of colorectal cancer on health systems.

Secondly, this manuscript highlights how AI can elevate the accuracy of polyp characterisation when used by regular endoscopists to nearly that of highly trained expert interventionalists. This development holds significant promise for overburdened healthcare systems worldwide, where access to specialist interventionalists is often limited. AI’s ability to assist in precise polyp characterisation can help mitigate the risk of misdiagnoses, reducing unnecessary treatments, and enhancing patient care quality. Moreover, by empowering non-experts with advanced AI tools, we can ensure that patients in underserved regions receive comprehensive care, irrespective of the available expertise.

In a world where resource limitations persist and not everyone has access to highly trained interventionalists, this manuscript’s findings underscore the profound public health implications of AI in colonoscopy. AI’s capacity to augment both adenoma detection and polyp characterisation in the hands of all proceduralists not only promises to enhance healthcare consistency but also signifies a crucial step towards healthcare equity. As we continue to harness the power of artificial intelligence in medicine, the potential to democratise expertise and improve the overall health outcomes of diverse populations becomes increasingly tangible and vital.

## Figures and Tables

**Table 1 cancers-15-05126-t001:** Commercially available artificial intelligence systems in colonoscopy.

Name	Company	Technique	Commercial Approval
EndoBRAIN	Cybernet Systems Corporation (Tokyo, Japan)	CADx	2018
GI Genius	Medtronic (Dublin, Ireland)	CADe	2019
EndoBRAIN-EYE	Cybernet Systems Corporation (Tokyo, Japan)	CADe	2020
DISCOVERY	Pentax Medical Company (Tokyo, Japan)	CADe	2020
ENDO-AID	Olympus Corporation (Tokyo, Japan)	CADe	2020
CAD EYE	Fujifilm (Tokyo, Japan)	CADe, CADx	2020
Wise Vision	NEC Corporation (Tokyo, Japan)	CADe	2020
EndoScreener	Wision A.I. (Shanghai, China)	CADe	2021

**Table 2 cancers-15-05126-t002:** Randomised controlled trials comparing artificial-intelligence-aided colonoscopy with control groups for adenoma detection.

Author, Year	CADe System	Control	Patients (*n*)	ADR (AI vs. Control)	Advanced ADR (AI vs. Control)
Nakashima et al., 2023 [11]	CAD EYE	HD-WLI	415	59.4% vs. 47.6% (*p* = 0.018)	7.2% vs. 7.7% (*p* = 1)
Xu et al., 2023 [12]	Eagle-Eye	HD-WLI	3059	39.9% vs. 32.4% (*p* < 0.001)	6.6% vs. 4.9% (*p* = 0.041)
Wang et al., 2023 [13]	EndoScreener	HD-WLI with second observer	1261	25.8% vs. 24.0% (*p* = 0.464)	0.314% vs. 0.39% (*p* = 0.562)
Wei et al., 2023 [14]	EndoVigilant	HD-WLI	769	35.9% vs. 37.2% (*p* = 0.774)	N/A
Ahmad et al., 2022 [15]	GI Genius	HD-WLI	658	71.4% vs. 65.4% (*p* = 0.09)	N/A
Gimeno-Garcia et al., 2022 [16]	ENDO-AID	HD-WLI	370	55.1% vs. 43.8% (*p* = 0.029)	11.6% vs. 12.1% (*p* = 0.89)
Repici et al., 2022 [17]	GI Genius	HD-WLI	660	53.3% vs. 44.5% (*p* < 0.02)	12.7% vs. 12.7% (*p* = 0.956)
Rondonotti et al., 2022 [18]	CAD EYE	HD-WLI	800	53.6% vs. 45.3% (RR 1.18, 95% CI 1.03–1.36)	18.5% vs. 15.9% (RR 1.03, 95% CI 0.96–1.09)
Shaukat et al., 2022 [19]	SKOUT	HD-WLI	1359	47.8% vs. 43.9% (*p* = 0.065)	N/A
Luo et al., 2021 [20]	Xiamen Innovision	HD-WLI	150	PDR 38.7% vs. 34.0% (*p* < 0.001)	N/A
Xu et al., 2021 [21]	N/A	HD-WLI	2352	PDR 38.8% vs. 36.2% (*p* = 0.183)	N/A
Liu P et al., 2020 [22]	EndoScreener	HD-WLI	790	29.01% vs. 20.91% (*p* = 0.009)	1.43% vs. 3.92% (*p* = 0.607)
Liu W et al., 2020 [23]	Henan Xuanweitang Medical Information Technology Co.	HD-WLI	1026	39.1% vs. 23.89% (*p* < 0.001)	2.88% vs. 6.45% (*p* = 0.821)
Repici et al., 2020 [24]	GI-Genius	HD-WLI	685	54.8% vs. 40.4% (RR 1.30, 95% 1.14–1.45)	10.3% vs. 7.3% (*p* = 0.769)
Wang et al., 2019 [10]	EndoScreener	HD-WLI	1058	29.12% vs. 20.34% (*p* < 0.001)	3.41% vs. 5.95% (*p* = 0.803)

**Table 3 cancers-15-05126-t003:** Meta-analyses comparing artificial-intelligence-aided colonoscopy with control groups for adenoma detection.

Author, Year	Studies (*n*)	Patients (*n*)	ADR (AI vs. Control)	≤5 mm Adenomas	≥10 mm Adenomas	Notes
Huang et al., 2022 [25]	10	6629	RR 1.43, *p* < 0.001	RR 1.71, *p* < 0.001	RR 1.73, *p* < 0.001	SSL per colonoscopy RR 1.53, *p* < 0.001
Sivananthan et al., 2022 [26]	7	5217	33.65% vs. 22.85%	0.691 adenomas per colonoscopy vs. 0.373 (pooled effect size 0.3, 95% CI 0.19–0.42)	N/A	91.7% higher detection of non-pedunculated adenomas
Ashat et al., 2021 [27]	6	5058	33.7% vs. 22.9% (OR 1.76, 95% CI 1.55–2.00)	OR 2.07, 95% CI 1.81–2.36, *p* < 0.001	OR 1.79, 95% CI 1.27–2.53, *p* < 0.001	
Barua et al., 2021 [28]	5	4311	29.6% vs. 19.3% (RR 1.52, 95% CI 1.31–1.77)	Mean difference, 0.15 (95% CI 0.12–0.28)	Mean difference 0.01, 95% CI 0.00–0.02)	
Deliwala et al., 2021 [29]	6	4996	OR 1.77 (95% CI 1.57–2.08)	OR 1.33 (95% CI 1.12–1.59)	OR 1.24 (95% CI 0.87–1.78)	
Hassan et al., 2021 [30]	5	4354	36.6% vs. 25.2%, RR 1.44 (95% CI 1.27–1.62)	RR 1.69 (95% CI 1.48–1.84)	RR 1.46 (95% CI 1.04–2.06)	SSL per colonoscopy RR 1.52 (95% CI 1.14–2.02)
Li et al., 2021 [31]	5	4311	OR 1.75 (95% CI 1.52–2.01)	N/A	N/A	
Nazarian et al., 2021 [32]	8	5577	OR 1.53 (95% CI 1.32–1.77)	N/A	N/A	
Spadaccini et al., 2021 [33]	6	4996	OR 1.78 (95% CI 1.44–2.18)	N/A	OR 1.69 (95% CI 1.10–2.60)	No difference in SSL detection, OR 1.37 (95% CI 0.65–2.88)
Zhang et al., 2021 [34]	7	5427	OR 1.72 (95% CI 1.52–1.95)	OR 1.42 (95% CI 1.18–1.72)	OR 0.71 (95% CI 0.46–1.10)	Less advanced adenomas (OR 0.70, 95% CI 0.50–0.97) SSL OR 0.87 (95% CI 0.61–1.23)
Aziz et al., 2020 [35]	3	2815	32.9% vs. 20.8%, RR 1.58 (95% CI 1.39–1.80)	N/A	N/A	

N/A= variable not reported.

**Table 4 cancers-15-05126-t004:** Tandem colonoscopies randomised to CADe or HD-WLI first.

Author, Year	Patients (*n*)	Adenoma Miss Rate (CADe vs. HD-WLI)	SSL Miss Rate (CADe vs. HD-WLI)	Non-Polypoid Adenoma Miss Rate	Right Colon Adenoma Miss Rate
Glissen-Brown et al., 2022 [41]	234	20.12% vs. 31.25% (*p* = 0.0247)	7.14% vs. 42.11% (*p* = 0.0482)	17.65% for CADe vs. 22.22% for HD-WLI (*p* = 0.5872)	Higher miss rate for HD-WLI in the right colon on multivariable analysis (OR 1.7865, *p* = 0.0436)
Wallace et al., 2022 [42]	230	15.5% vs. 32.4% (*p* < 0.001)	0% vs. 33.33% (*p* = 0.455)	Lower miss rate with CADe for nonpolypoid adenomas (OR 0.34, *p* < 0.001)	18.3% with CADe vs. 32.53% with HD-WLI (*p* = 0.004)
Kamba et al., 2021 [43]	346	13.8% vs. 36.7% (*p* < 0.001)	13% vs. 38.5% (*p* = 0.0332)	13.38% for CADe vs. 45.26% for HD-WLI (*p* < 0.001)	9.23% for CADe vs. 44.05% for HD-WLI (*p* < 0.001)

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
