# Peer review of "The Role of Artificial Intelligence in Colorectal Cancer Screening: Lesion Detection and Lesion Characterization"

_cancers, 2023, doi:10.3390/cancers15215126_

Round 1

Reviewer 1 Report

This manuscript systemically reviewed the potential applications of AI in colorectal cancer screening. The literatures cited are comprehensive. One minor error is the summary. Both line 245 and line 395 are titled as summary. They need to be consolidated.  

Author Response

Dear Reviewer, 

Thank you for taking the time to review our manuscript. We greatly appreciate your review and feedback. The two 'summary' paragraphs are each summarising the major sections of the paper, the first being the summary for polyp detection, the second for polyp characterisation. Hopefully with appropriate formatting at the time of publication this will be more clearly represented. 

Thank you again for your feedback.

Edward Young

Reviewer 2 Report

The paper may hold the potential to offer extra contextual details regarding the fundamental ideas and principles of artificial intelligence (AI), such as machine learning, deep learning, neural networks, and computer-aided detection and diagnosis. By avoiding doing so, it would hinder a better understanding of the article and its implications for readers who have limited familiarity with artificial intelligence. Furthermore, the paper would benefit from the utilization of consistent and precise terminology when discussing the various types of AI systems in colonoscopy, such as CADe, CADx, EndoBRAIN, GI Genius, etc. Some of these terms remain inadequately defined or explained within the paper, potentially leading to confusion or ambiguity for readers. To tackle this problem, it is suggested that the document offers a more extensive examination and conversation about the limitations and difficulties connected with AI systems in colonoscopy. This should encompass ethical, legal, social, and economic considerations, along with the necessity for additional validation, standardization, and regulation of these technologies. By implementing these components, the document would provide a more comprehensive and well-rounded viewpoint on the function of AI in colorectal cancer screening.

The document is skillfully composed, well-arranged, and meticulously structured, featuring lucid headings, subheadings, tables, and figures. The document adheres to the prescribed guidelines of the esteemed Digestive Endoscopy journal and employs an appropriate style of citation. The document showcases a commendable command of the English language, employing fluid and coherent discourse, precise grammar and spelling, as well as apt vocabulary and terminology. The document employs a formal and scholarly tone, shunning colloquialisms and jargon, and upholds consistency and clarity throughout. The document could benefit from addressing minor concerns, such as avoiding the repetition of certain phrases or words, such as "AI" or "CADe," by utilizing synonyms or pronouns when feasible. Moreover, providing additional details or explanations for certain abbreviations or acronyms, such as "SSLs" or "AMR," particularly when they are initially introduced or utilized in distinct contexts, would enhance the document's quality. Furthermore, it is advisable to scrutinize for any typographical errors or punctuation mistakes, like "Endosc Int Open 2021;9:E1004-e1011." or "Gastroenterology 2020;158:2169-2179 e8." Additionally, ensuring the completeness and uniformity of all references, such as "Am J Gastroenterol 2020;115:138-144." or "Cancers (Basel) 2021;13," is crucial.

Author Response

Dear Reviewer,

Thank you very much for taking the time to review our manuscript in such detail, and for your feedback below.

We have added additional contextual details regarding the principles of AI in gastroenterology and designated an additional section of the manuscript to this. This component of the manuscript does remain brief in comparison, although we felt that the focus of the manuscript was on the clinical application of AI and therefore aimed to keep the intricacies of AI brief in order to not overcomplicate the manuscript further.

We have attempted to further define terms and abbreviations in order to improve the manuscript's clarity as described. 

We have aimed to reduce repetition of terms as described below by exchanging them with synonyms or alternative descriptors. 

We thank you again for highlighting the errors in the references and have been through the reference list in detail to fix these.

Thank you again for your detailed review of our manuscript. We look forward to your review of our revision.

Kind regard,

Edward Young
